# Numerical Analysis, Optimization, and Multi-Criteria Design of Vacuum Insulated Glass Composite Panels

**DOI:** 10.3390/ma16134722

**Published:** 2023-06-29

**Authors:** Izabela Kowalczyk, Damian Kozanecki, Sylwia Krasoń, Martyna Rabenda, Łukasz Domagalski, Artur Wirowski

**Affiliations:** 1Department of Structural Mechanics, Lodz University of Technology, Politechniki 6, 93-590 Lodz, Poland; izabela.kowalczyk@dokt.p.lodz.pl (I.K.); damian.kozanecki@dokt.p.lodz.pl (D.K.); lukasz.domagalski@p.lodz.pl (Ł.D.); 2Department of Concrete Structures, Lodz University of Technology, Politechniki 6, 93-590 Lodz, Poland; martyna.rabenda@p.lodz.pl

**Keywords:** composite panels, finite element method, dynamic analysis, fractals, VIG

## Abstract

The subject of this study is Vacuum Insulated Glass (VIG) panels, which consist of two glass panes with an evacuated space and evenly distributed micro-support pillars between them. The deflection of panes towards the centre of the structure caused by atmospheric pressure is a mechanical problem that occurs in this type of structure. The aim of this study was to extend previous research on the optimal arrangement of support pillars in terms of eigenfrequencies and dynamics to include aesthetic aspects. Using Abaqus/CAE v2017 software, a large number of numerical models were created and subjected to a comprehensive multi-criteria analysis. Fractal analysis was employed to automatically assess the aesthetics of the proposed solutions. The study presents theoretical solutions that could be implemented in industrial production. The presented study shows that it is possible to effectively extend the criteria for optimizing the arrangement of pillars with new design criteria. Most studies focus on pillar placement, amount, or shape in terms of panes thermal or mechanical properties. Due to the increasing number of VIG panels applications in places exposed to external vibrations, other design criteria for VIG panels are also required and are provided by the following study.

## 1. Introduction

### 1.1. VIG Panels

Nowadays, energy management is an important problem in the world economy. Its price has risen sharply in recent years. It is caused by the need to move away from fossil fuels and the dwindling resources of the planet. The recent geopolitical problems are an additional factor deepening the upward trend in energy prices. Therefore, any initiative aimed at saving energy is a priority for the economy.

The subject of this work is VIG (Vacuum Insulating Glass) panels. They represent the latest technological achievement that is already implemented in many countries. The VIG technology consists of two high-strength glass panes with a vacuum between them (Figure 1). In order to counteract the atmospheric pressure and the contact of the panels, they are connected to micro-support pillars—elements usually made of steel with a diameter of 0.3–0.5 mm [1]. Moreover, the panes are sealed around the edges. Consequently, this design provides excellent thermal properties to VIG panels. Their U-value can be as low as 0.2–0.4 W/m^2^K [2].

### 1.2. Fractals in Engineering

In recent years, scientists and engineers have been looking for inspiration to create new, original structures with better properties. Among the many interesting trends, the attempts to apply structures called fractals in various fields of technology are remarkable [3,4,5]. These are mathematical structures discovered and described by Mandelbrot in the 1970s. Their representation in nature is significant [6]. They are characterized by a sophisticated, self-similar structure, the similarity scale of which is described by a parameter called fractal dimension. Over the next years, fractals will be subject of the extensive research. The individual section of mathematics, such as fractal geometry, was created. The wide possibilities of fractal applications were also analysed, both to create new technical structures and to use the so-called fractal analysis to describe the features of existing structures [7]. Fractals application in architecture and civil engineering [8] deserves special attention—particularly in the dynamics of plate structures [9]. Fractals have been seen in architecture across many eras and cultures.

The richness and variety of fractal forms naturally occurs in nature but also in the architecture of many cultures. This leads to natural questions of the aesthetic value of fractals. The proportions related to the similarity of fractal structures provides the observer an aesthetic feeling which has already been the subject of scientific research [10]. The mathematical description of fractal structures gives the engineers possibility to quantify the aesthetic criterion that can be used for generative design of building objects [11].

### 1.3. Motivation

Thermal properties of VIG panels [12,13], strength properties of support-pillars and their static properties [14,15] were subject of numerous studies. However, in recent years VIG technology has been analysed also in terms of its acoustic properties [16].

An interesting trend in the recent research on VIG technology are studies on its dynamic properties. In [17], the authors, using an advanced FEM model, analysed a number of plates with different geometries and compared their dynamic properties. By determining the value of the natural frequency of these structures, it is possible to determine the possibility of using VIG plates in buildings exposed to external dynamic factors, such as traffic or industrial vibrations. Another work worth mentioning is [18], in which the authors made an attempt to optimize pillar distribution in terms of the dynamic properties of the entire structure. They used an analytical model based on the analysis of classical equations of motion of thin plates elastically connected with connectors. Then, using genetic algorithms and applying dynamic optimization criteria, they obtained non-trivial topological solutions for the structure of VIG plates.

The motivation to undertake this work was the willingness to summarize and also continue the research contained in [18]. It is worth emphasizing that authors use a completely different theoretical model of VIG plates. Combining, comparing and analysing advanced FEM models with a simple analytical approach allows the selection of models of this type of structure that is easier to apply in the future.

Another motivation to start this work were the conclusions from the work [18]. The authors, after analysing the dynamic properties of many VIG-type structures, concluded that the optimization used criteria require further refinement and extension, as they slightly affect the distribution of pillars. Hence, the need to extend the optimization criterion with new aspects, which is undoubtedly the fractal analysis. By determining the correlation coefficient of a given structure with the mathematical structure of the fractal, it is possible to evaluate the aesthetic properties of the arrangement of pillars in the VIG panels. This would enable a more comprehensive approach to the optimization of these types of structures.

### 1.4. Purpose of the Work

The first aim of this work is to compare the accuracy of various numerical models of VIG plates in terms of dynamics. Demonstrated models are simple and easier to apply. They provide correct and accurate results in relation to advanced three-dimensional structure models. They allow for faster computations, which is important in the case of genetic calculations and optimization.

The second important goal of the work is the use of multi-criteria optimization to design the arrangement of connectors in VIG panels. By extending the objective function with a parameter that determines the degree of similarity of a given structure to a fractal, a simple quantitative parameter characterizing the aesthetic properties of panels was obtained. The relationship between the aesthetic properties of the given structure and its fractal properties has been investigated by other authors for various problems [10]. The distribution of pillars, which in the case of structural elements intended for use in exposed places such as window partitions, is of key importance for the comfort of users. To determine a parameter characterizing the degree of similarity in the arrangement of pillars on the VIG plate, and thus to investigate its aesthetic properties, the mass-radius method was used [19].

## 2. Numerical Modelling

### 2.1. Geometry and Support Conditions

The main issue in selecting the geometry and support conditions was the need to enable compatibility between all analysed modelling methods, which differ significantly in their approach. Only by providing the same support conditions obtained results were comparable to each other. Ultimately, simple support conditions were assumed on all edges of the plate (i.e., all displacements were blocked and all rotations were released at all nodes—Figure 2).

In the case of the three-dimensional FEM model, the displacements were blocked only in the internal panes of both plates (Figure 3). In the case of blocking the displacements along the entire thickness of the glass elements, restraint would have appeared on a given edge [17].

Support of the middle panes of both glass plates was not considered as this would significantly increase the stiffness of the entire model. The thickness of the glass plates is much greater (6–8 mm) than the vacuum thickness (0.3 mm). This makes the arm of horizontal support forces much (about 20 times) greater than the arm of corresponding forces in the simplified model.

### 2.2. Simple FEM Model

A simple FEM model was made using the commercial engineering software Autodesk Robot Structural Analysis Professional v2022. A shell-member model was built, in which the glass plates were represented as flat shell elements simply supported in their middle panes. Pillars, represented as bar elements with six degrees of freedom in each node, were placed between plates (Figure 4). The division of the regular FEM mesh of both plates was adjusted so the mesh stands out at the point of the pillar’s connection.

### 2.3. 3D FEM Model

The model implemented in ABAQUS based on three-dimensional C3D8R elements: first order, reduced integration element with hourglass control activated was selected as the most advanced 3D model using FEM [17] (Figure 5).

The interpolation function for C3D8R element shown in Figure 5 is given by the equation:(1)$$u=N^{I}\left(g,h,r\right)u^{I}\;\mathrm{sum}\;\mathrm{on}\;I.$$
where ***I*** denote the node of the element. The last four vectors $\Gamma_{\alpha }^{I}$, α = 1, 2, 3, 4 are the hourglass base vectors. The gradient matrix $B^{I}$ is defined by integrating over the element:(2)$$N^{I}\left(g,h,r\right)=\frac{1}{8}\Sigma^{I}+\frac{1}{4}g\Lambda_{1}^{I}+\frac{1}{4}h\Lambda_{2}^{I}+\frac{1}{4}r\Lambda_{3}^{I}+\frac{1}{2}hr\Gamma_{1}^{I}+\frac{1}{2}gr\Gamma_{2}^{I}+\frac{1}{2}gh\Gamma_{3}^{I}+\frac{1}{2}ghr\Gamma_{4}^{I},$$

The shape functions are the same as for the C3D8 element and can be found in [20] (2):(3)$$B_{i}^{I}=\frac{1}{V_{el}}\int_{V_{el}}\;N_{i}^{I}\left(g,h,r\right)dV_{el},\;N_{i}^{I}\left(g,h,r\right)=\frac{\partial N^{I}}{\partial x_{i}},$$
where $V_{el}$ is volume of the element and *I =* 1, 2, 3. In the centroidal strain formulation the gradient matrix is simply given:(4)$$B_{i}^{I}=N_{i}^{I}\left(0,0,0\right).$$

Considering the above, it may be seen that centroidal strain formulation reduces the amount of effort required to compute the gradient matrix. In ABAQUS the artificial stiffness method and the artificial damping method showed in [21] are used to control the hourglass modes in these elements. In the work [22], the effectiveness of C3D8R elements with hourglass control was compared to other types of elements.

### 2.4. Analytical Model

The analytical model included in this study was based on the assumptions made in [18]. Let *Ox_1_x_2_x_3_* be an orthogonal Cartesian coordinate system. Two plates of thicknesses *h_i_*, *i* = 1, 2 made of the same material (density *ρ*, Young’s modulus *E* and Poisson number *ν*) were considered. There are *n* springs, each of stiffness *k_j_*, located in points with coordinates $\left(x_{1,j},x_{2,j}\right),\;j=1,\ldots ,n$ are located between those plates (Figure 6).

Due to the small thickness of the glass plates and their slight deformations, the classical theory of thin plates, known as the Kirchhoff—Love theory, was applied. According to the studies conducted by the authors in [15], the results obtained using the linear and classical theory for frequency values are fully realistic. According to the theory of thin plates, the displacement takes the form of:(5)$$w_{\alpha }\left(x\right)=w_{\alpha }^{0}\left(x_{1},x_{2}\right)-x_{3}\frac{\partial w^{0}}{\partial x_{\alpha }}=w_{\alpha }^{0}-x_{3}w_{,\alpha }^{0};\;\;\;\;\;\;\alpha =1,2\;w_{3}\left(x\right)=w^{0}\left(x_{1},x_{2}\right)$$
strain-displacement relations are presented as:(6)$$\varepsilon_{\alpha \beta }=\frac{1}{2}\left(w_{\alpha ,\beta }^{0}+w_{\beta ,\alpha }^{0}+w_{,\alpha }^{0}w_{,\beta }^{0}\right)-x_{3}w_{,\alpha }^{0}\;\varepsilon_{\alpha 3}=0;\;\varepsilon_{33}=0.$$

The action functional of the problem is given by:(7)$$A=\overset{t_{2}}{\underset{t_{1}}{\int }}\left(W-K\right)dt,$$
where the strain energy is equal to:(8)$$W=\frac{1}{2}\underset{A}{\int }\left(\overset{2}{\underset{i=1}{\sum }}\partial_{\alpha \beta }w_{i}D^{i}{}_{\alpha \beta \gamma \omega }\partial_{\gamma \omega }w_{i}\right)dA+\frac{1}{2}\overset{n}{\underset{j=1}{\sum }}k_{j}{\left[w_{2}\left(x_{1,j},x_{2,j},t\right)-w_{1}\left(x_{1,j},x_{2,j},t\right)\right]}^{2},$$
and kinetic energy is given by:(9)$$K=\frac{1}{2}\underset{A}{\int }\left(\overset{2}{\underset{i=1}{\sum }}\mu_{i}{\dot{w}}_{i}^{2}\right)dA.$$

Let $w_{i}=w_{i}\left(x_{1},x_{2},t\right)$ be the transverse deflection; overdot stands for the derivative with respect to time $t=t_{0},t_{1}$. The coefficients $D_{\alpha \beta \gamma \delta }^{i}$ represent the plate stiffness tensor elements and are given by the following formula:(10)$$D_{\alpha \beta \gamma \omega }^{i}=D_{0}^{i}\left[v\delta_{\alpha \beta }\delta_{\gamma \omega }+\frac{1-v}{2}\left(\delta_{\alpha \gamma }\delta_{\beta \omega }+\delta_{\alpha \omega }\delta_{\beta \gamma }\right)\right],$$
where the plate flexural rigidity is equal to:(11)$$D_{0}^{i}=\frac{Eh_{i}^{3}}{12\left(1\pm v^{2}\right)}.$$

The derivatives are given as follows:(12)$$\partial_{\alpha }w_{i}\equiv \frac{\partial w_{i}}{\partial x_{\alpha }},\;\;\partial_{\alpha }w_{i}\equiv \frac{\partial^{2}w_{i}}{\partial x_{\alpha }\partial x_{\beta }},\;\;\;{\dot{w}}_{i}=\frac{\partial w_{i}}{\partial t}.$$

Coefficient $\partial_{\alpha \beta }$ stands for Kronecker delta. The indices α,β,γ,ω are related to the *Ox*_1_*x*_2_ coordinate system and run through 1, 2.

### 2.5. Natural and Forced Vibrations

In this study, the dynamic properties of VIG plates were investigated in terms of natural vibrations in which various approaches to numerical modelling were compared and in terms of forced vibrations. The eigenproblem can be expressed in a standard form:(13)$$\left(K-\omega^{2}M\right)u=0,$$
where **K**, **M** and **u** stand for matrix of stiffness, matrix of mass and vector of displacement, respectively. The displacement vector is equal to:(14)$$u=\left[a_{1}^{1}a_{1}^{2}\;\ldots a_{1}^{m}a_{2}^{1}a_{2}^{1}\;\ldots a_{2}^{m}\right].$$

Elements of matrices **K** and **M** are, respectively, given by following formulas:(15)$$K_{i,j}=\frac{\partial^{2}W}{\partial u_{i}\partial u_{j}},$$
(16)$$M_{i,j}=\frac{\partial^{2}K}{\partial u_{i}\partial u_{j}}.$$

Considering forced damped vibrations the system of equations is assumed as:(17)$$\mathrm{Ku}+C\dot{u}+M\ddot{u}=Q\;\cos\left(\Omega t\right).$$
where **K**, **C**, **M** stand for stiffness, damping and mass matrices, respectively, **Q** is load, **u** is displacement vector Ω is forcing frequency and *t* is time. The damping matrix has been defined so that the damping could be considered as slight:(18)$$C=\alpha {\left(\mathrm{MK}\right)}^{\frac{1}{2}},$$
where *α* is assumed equal to 10^−2^ [18]. The formula describing a uniformly distributed load is assumed as:(19)$$\;Q\left(x_{1},x_{2},t\right)=\sum_{j=1}^{m}q^{j}\varphi_{j}\left(x_{1},x_{2}\right)cos\left(\Omega t\right),$$
where *q^j^* are known coefficients dependent on the loading case.

Application of the multimodal approach leads to the following solution:(20)$${u=a}_{C}cos\left(\Omega t\right)+a_{S}sin\left(\Omega t\right).$$

Coefficients **a***_C_* and **a***_S_* are defined as:(21)$$a_{C}={\left\{\left(K-\Omega^{2}M\right)+\Omega^{2}C\left[{\left(K-\Omega^{2}M\right)}^{-1}\right]\right\}}^{-1}Q,$$
(22)$$a_{S}=\Omega {\left(K-\omega^{2}M\right)}^{-1}C{\left\{\left(K-\Omega^{2}M\right)+\Omega^{2}C\left[{\left(K-\Omega^{2}M\right)}^{-1}\right]\right\}}^{-1}Q.$$

### 2.6. Simple Genetic Algorithm (SGA)

SGA (simple genetic algorithm) was selected as the tool for topology optimization of the plate. A single individual (VIG plate) was encoded by genotype in the form of an array with values 1 or 0 representing the existence or absence of a pilaster at a given position in the plate [18]. An example of coded pillars arrangement is shown in Figure 7.

Other SGA parameters, such as the selection of an appropriate method of interbreeding of individuals taking into account the characteristics of the problem, are described in detail in [9].

## 3. Optimization Criteria and Fractal Analysis

In this paper, several optimization criteria of various types have been defined. The first criterion used was the dynamic one, defined analogously to the work [18], i.e., as the average of the ratio of the maximum deflection of the lower and upper plate in the frequency range from 220 Hz to 440 Hz:(23)$$f=average\left(\frac{max\left|w_{2}\right|}{max\left|w_{1}\right|}\right).$$

The function (23) for a solid plate (i.e., a plate with evenly spaced pillars) is not less than 0.985. However, as shown in [18], limiting ourselves to this criterion gives only a small possibility of topological optimization of the joint arrangement.

Fractal analysis was used as another criterion for advanced optimization of pillar distribution. It determines the degree of similarity of a given structure to the theoretical fractal structure. There are many methods of determining the so-called fractal dimension of the structure. Due to the point nature of the pillars distribution decision to use an algorithm for the objective function calculation based on the simplified mass-radius method described in [19] was made. This algorithm was implemented in two steps:The so-called spinning radius *R* (Figure 8) varying from 0 to the maximum value depending on the size of the given plate, and for each *R*, the number of pilasters in total *M*(*R*) was counted in circles with centres in all pillars of a given spinning radius *R*;Because for an ideal fractal structure there is a relationship [19]:
(24)$$M\left(R\right)\sim R^{D},$$
where *D* is the so-called fractal dimension.

In order to investigate the degree of similarity between VIG plate structure and the ideal fractal a logarithmic approximation of *M*(*R*) data series was used. Then, *r* coefficient was determined which defines the degree of adaptation the logarithmic curve to the data described in the previous step. Classical formulas for the least squares approximation were used:(25)$$r=\frac{n\sum_{i=1}^{n}x_{i}y_{i}-\sum_{i=1}^{n}x_{i}\sum_{i=1}^{n}y_{i}}{\sqrt{n\sum_{i=1}^{n}x_{i}^{2}-{\left(\sum_{i=1}^{n}x_{i}\right)}^{2}}\sqrt{n\sum_{i=1}^{n}y_{i}^{2}-{\left(\sum_{i=1}^{n}y_{i}\right)}^{2}}},$$
where $x_{i}=\log\left(R\right)$ and $y_{i}=\log\left[M\left(R\right)\right]$.

The correlation coefficient *r* obtained in (25) was used directly as an objective function. Its values are in the range from 0 to 1.

Equal weights and linear combination of the above objective functions were used for the multi-criteria analysis:(26)w=0.5f+0.5r.

The results obtained solely on the basis of the criterion defined by the Formula (21) were shown and analysed in [18]. This paper focuses on the results obtained for the criteria defined by the Formulas (23) and (24).

**Figure 8 materials-16-04722-f008:**
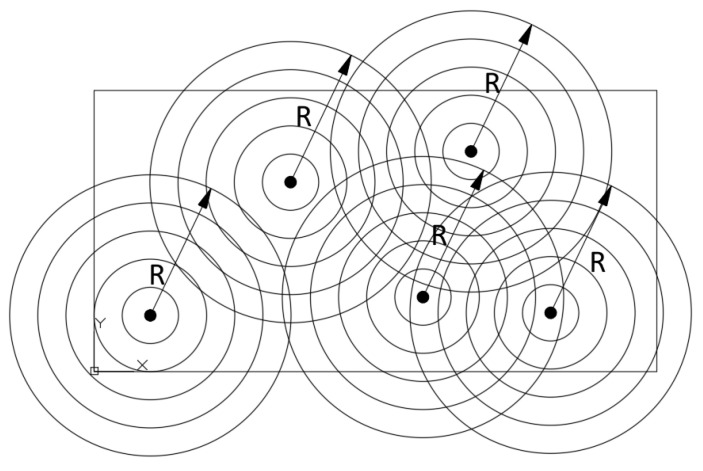
The idea of calculation of the fractal dimension of a structure based on the spinning radius [19].

## 4. Calculation Results

### 4.1. Comparison of Various Numerical Models

Comparative numerical calculations were performed for two pillars systems. In both cases, a plate with a glass pane thickness of *h* = 7.5 mm, vacuum width *h*_v_ = 0.3 mm and plate dimensions 2.0 m × 1.0 m was used. Glass with Young’s modulus *E* = 72 GPa, Poisson’s ratio *ν* = 0.33 and density ρ = 2500 kg/m^3^ was assumed. Support-pillars are 0.3 mm height and diameter. The material for pillars is steel, with Young modulus *E*_s_ = 200 GPa, the mass density *ρ*_s_ = 7850 kg/m^3^, and the Poisson’s ratio *ν*_s_ = 0.3. The following pillars systems were used (Figure 9):

For the analysis using FEM, in addition to the given material data, the following approaches were adopted:Simple FEM model—flat plate quadrilateral finite elements with a relatively sparse division of 50 mm.The 3D FEM model—three-dimensional finite elements were used. The glass panels were divided with a mesh size of 2.5 mm (0.5 mm near the support pillars), while the support pillars themselves were divided more accurately.

In the case 0375, the following results of the first several natural frequencies were obtained (Table 1).

The 0375X was analysed as the second case, which differed in more pillars in the centre of the slab. In the case of the 0375X, the following results of the first sixteen natural frequencies were obtained (Table 2).

When comparing the values of individual frequencies of natural vibrations, the modes of natural vibrations obtained in all models were analysed. They can be divided into two groups: vibrations in phase and vibrations in counter-phase. In the former case, both plates swing in the same direction at the same time (Figure 10a–d), in the latter case, plates swing in the opposite direction (Figure 10e–h).

It is worth noting that in the case of vibrations in the phase, the pilasters have no significant effect on the shape and the result of the natural vibrations values because both panes vibrate identically—the pillars do not undergo major deformations. However, in the case of vibrations in the counter-phase, the pillars undergo significant deformations, and their parameters and modelling method are of key importance for the final result. In general, the obtained results are in line with expectations. In most cases, the results obtained from all three models are consistent with each other, with differences not exceeding 5–6%.

However, in both cases, both 0375 and 0375X, the results obtained for models I and II (analytical model and FEM model) in the case of the lowest vibration frequencies significantly differ from the values obtained for model III (FEM 3D)—even by 24% (marked green in Table 1 and Table 2). The results for the 3D model are significantly higher. This is due to a completely different method of supporting these models, which is of greatest importance precisely for the first forms and frequency of vibrations. As the thicknesses of the glass plates is much greater than the thickness of the vacuum, a realistic 3D model is much stiffer than the corresponding 2D model. It was observed after analysing the fragment of the first mode of natural vibrations located in the support zone (Figure 11).

The form of deformation, in the case of 3D FEM model (Figure 10a), resembles a fixed scheme more than a simply supported scheme (Figure 10b).

For some cases of counter-phase vibrations large discrepancies in the results between the models were obtained, i.e., 3 (Figure 10e) and 10 (Figure 10g). Method of modelling and meshing of pillars may have had an influence on them. This problem was analysed in [17].

The results from the 0375 and 0375X models were also compared. It can be clearly seen that, in the case of the 0375X, the use of pillars in the centre of the slab completely eliminated the 3rd form of vibrations in the 0375 model (Figure 10e, Table 1 and Table 2).

### 4.2. Multi-Criteria Optimization

The rest of the numerical calculations for a number of cases which varied the dimensions of the VIG plate were performed (length *L*_1_, width *L*_2_ and the thickness of the glass plate h). Other geometrical and material parameters were assumed as in the previous analysis. The list of case numbers along with the corresponding combinations of dimensions are summarized in Table 3.

The analytical model of VIG plate and optimization using previously described genetic algorithms were used for the calculations. The following pillars distributions inside the VIG plate were obtained (Figure 12).

For square plates, the optimal forms are point-symmetrical with rotational symmetry of 90° angle (cases 0125 and 0175). As the disproportion between plate length and width increases, the optimal forms develop toward axisymmetric systems. An interesting relationship for the cases with a thicker glass plate (0175 and 0275) may be noticed. Due to the greater stiffness of the individual glass panes and the consequent different degree of energy transfer between the plates correlated with the objective function (21), the optimal forms aim at the ones shown in Figure 13.

In the case of using the fractal objective function defined by the Formula (23), the optimal solutions become more complicated. It is clear that trivial solutions are rejected. In the case of a square plate (0125), it is characterized by quasi-rotational symmetry with 90° angle. However, in the case of rectangular plates (0225 and 0325) the evolution of forms in relation to those shown in Figure 13 towards more differentiated solutions, but with the preservation of some areas of a linear arrangement of pillars.

## 5. Summary and Conclusions

As a result of the numerical calculations and analyses the following conclusions were formed:Natural frequencies analysis of VIG panels requires 3D model application, the results obtained from simplified models are significantly underestimated;VIG panels are characterized by two types of vibration: in phase (both glass panes bend in the same direction) and in counter-phase (both glass panes bend in opposite directions), pillars, their geometry and modelling method are of key importance for vibrations in counter-phase;It is possible to effectively extend the criteria for optimizing the arrangement of connectors with new design criteria;Fractal analysis can be a tool for VIG panels design.

In further research, the authors intend to focus on refining the programming genetic algorithm and determining the fractal size of the structure, so that it is possible to significantly tighten the pilaster mesh, as well perform a detailed analysis of pillars parameters on the values and forms of natural vibrations.

## Figures and Tables

**Figure 1 materials-16-04722-f001:**
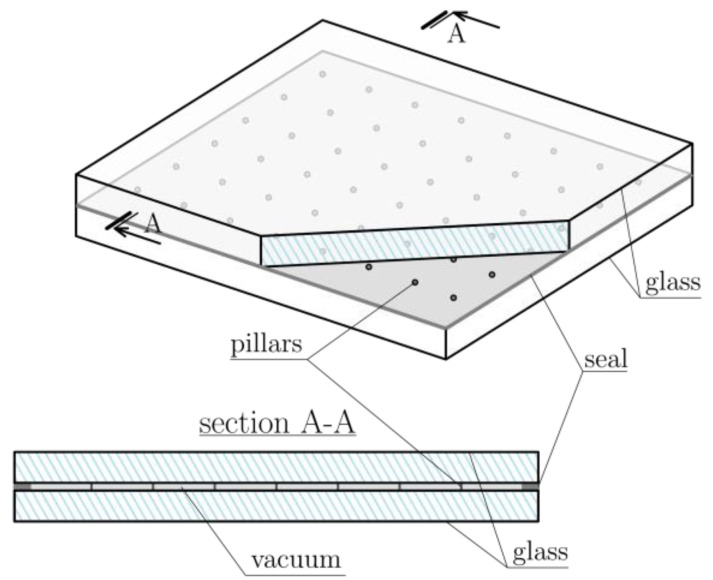
Schematic diagram of a vacuum glazing.

**Figure 2 materials-16-04722-f002:**
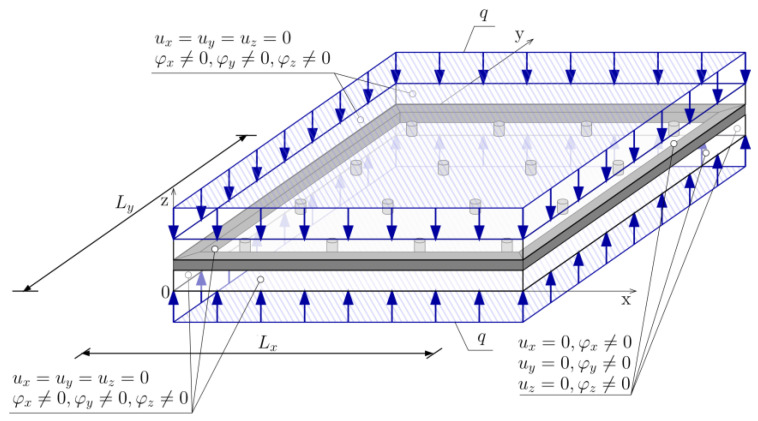
VIG plate computational model—geometry.

**Figure 3 materials-16-04722-f003:**
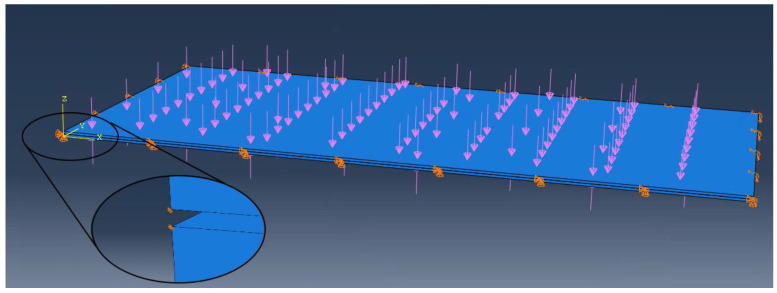
The method of support used in the FEM 3D model.

**Figure 4 materials-16-04722-f004:**
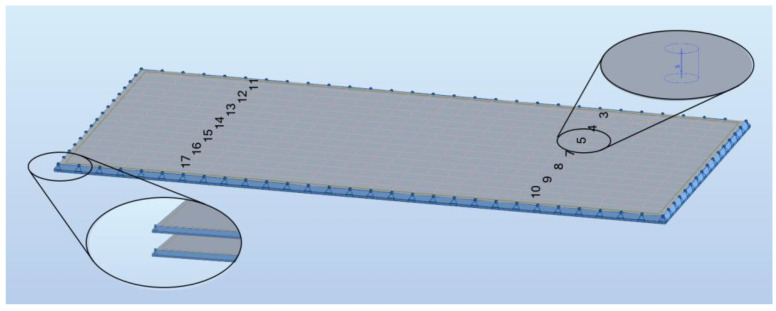
A Shell-member model created in Robot Structural Analysis Professional.

**Figure 5 materials-16-04722-f005:**
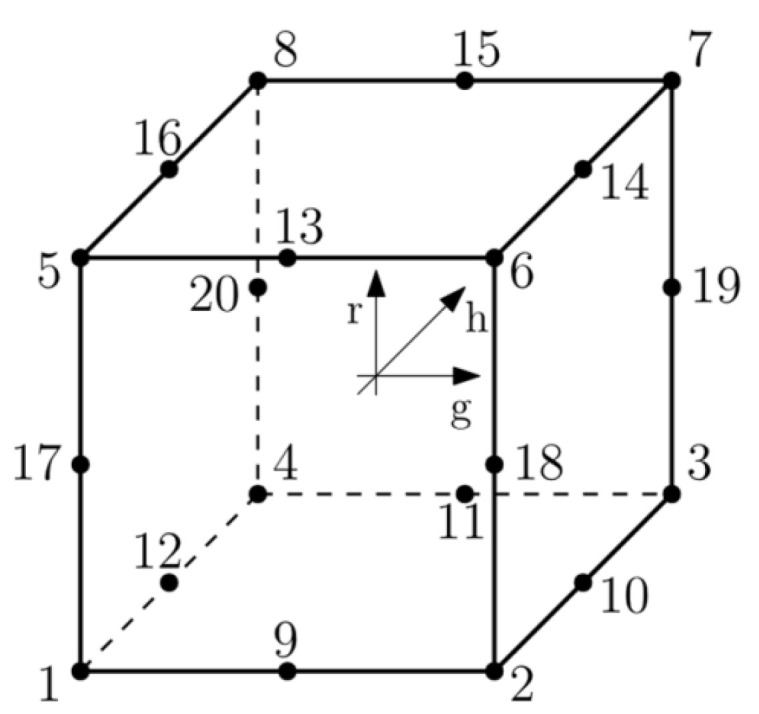
C3D8R element designations. Numbers stand for nodes.

**Figure 6 materials-16-04722-f006:**
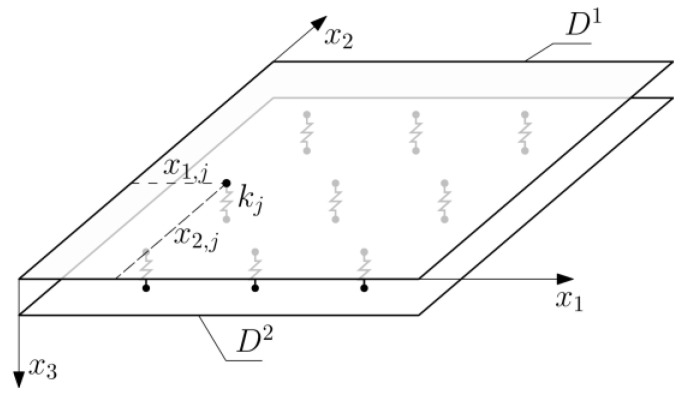
Computational model of the plate.

**Figure 7 materials-16-04722-f007:**
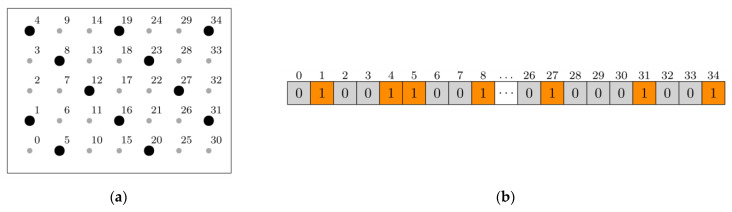
Example of chromosome coding: (**a**) schematic plate with pillars (black point—pillar occurs, grey point—pillar lacking), (**b**) part of chromosome representing an individual plate.

**Figure 9 materials-16-04722-f009:**
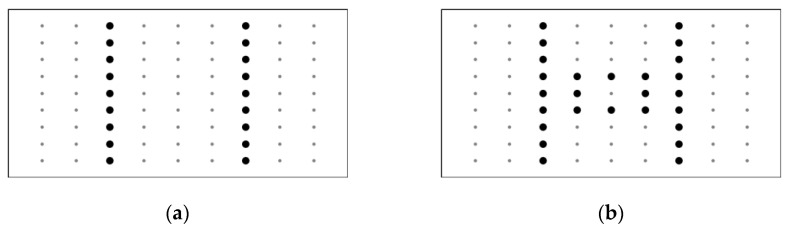
Applied pillars arrangement in two calculation examples: (**a**) case number 0375, (**b**) case number 0375X. Grey dots stand for the lack of the pillars and black stand for pillars.

**Figure 10 materials-16-04722-f010:**
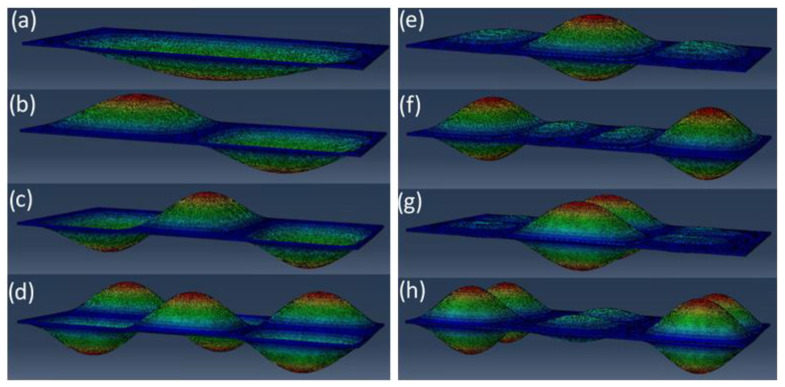
Selected z-normalized wane vibration modes for plate 0375: (**a**–**d**) vibration in phase, (**e**–**h**) vibration in counter phase; (**a**) 1st mode shape, (**b**) 2nd mode shape, (**c**) 4th mode shape, (**d**) 11th mode shape, (**e**) 3rd mode shape, (**f**) 6th mode shape, (**g**) 10th mode shape, (**h**) 14th mode shape.

**Figure 11 materials-16-04722-f011:**
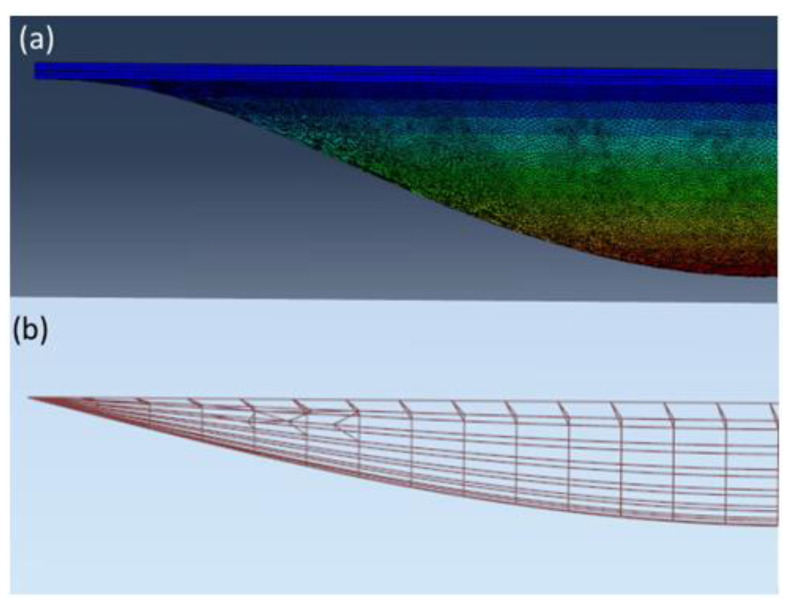
Fragment of the first z-normalized wane mode of natural vibrations for case 0375 in two calculation models (**a**) 3D FEM model; (**b**) simple FEM model.

**Figure 12 materials-16-04722-f012:**
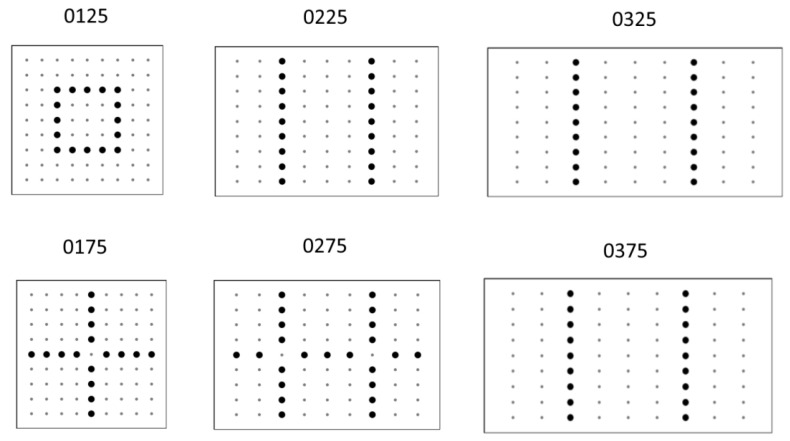
Optimal pillars distributions for the mixed criterion for different cases of plate geometry. Grey dots stand for the lack of the pillars and black stand for pillars.

**Figure 13 materials-16-04722-f013:**
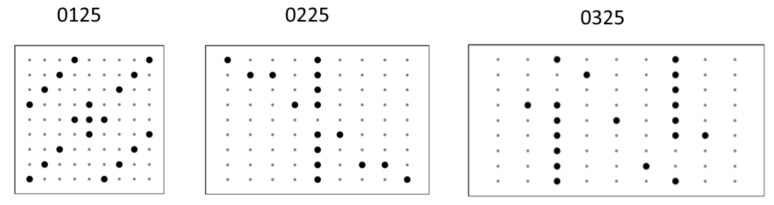
Optimal pillars distributions for the fractal criterion for different cases of plate geometry. Grey dots stand for the lack of the pillars and black stand for pillars.

**Table 1 materials-16-04722-t001:** The first eighteen eigenfrequencies calculated in three different numerical models for the case number 0375. The frequencies that are most influenced by the problem of mapping with the method of supporting the 3D model are marked in green. Vibrations in counter-phase are marked in blue.

Shape Number	I. Analytical, (Hz)	II. Simple FEM, (Hz)	III. 3D FEM, (Hz)	(I–II)/II, (%)	(II–III)/III, (%)	Figure Number, [-]
1	24.20	23.93	31.56	−1	24	10a
2	38.72	38.20	43.53	−1	12	10b
3	58.18	39.26	56.30	−48	30	10e
4	62.91	61.97	65.38	−2	5	10c
5	82.27	81.39	77.94	−1	−4	-
6	83.53	85.49	83.99	2	−2	10f
7	-	-	92.66	-	-	-
8	96.79	95.24	94.43	−2	−1	-
9	96.79	95.34	95.92	−2	1	-
10	94.16	93.59	107.46	−1	13	10g
11	120.99	118.59	119.04	−2	0	10d
12	139.23	138.00	130.76	−1	−6	-
13	-	-	136.29	-	-	-
14	140.35	139.41	139.48	−1	0	10h
15	145.59	151.14	149.69	4	−1	-
16	154.86	155.23	149.84	0	−4	-
17	169.41	177.29	171.85	4	−3	-
18	-	-	176.64	-	-	-

**Table 2 materials-16-04722-t002:** The first sixteen eigenfrequencies calculated in three different numerical models for the case number 0375X. The frequencies that are most influenced by the problem of mapping with the method of supporting the 3D model are marked in green. Vibrations in counter-phase are marked in blue.

**Shape** **Number**	**I. Analytical,** ** (Hz)**	**II. Simple FEM,** **(Hz)**	**III. 3D FEM,** **(Hz)**	**(I–II)/II,** **(%)**	**(II–III)/III,** **(%)**
1	24.20	23.93	31.65	−1	24
2	38.72	38.20	44.36	−1	14
3	62.91	61.97	66.02	−2	6
4	82.27	81.39	81.31	−1	0
5	89.69	-	89.13	-	-
6	90.32	-	89.71	-	-
7	96.79	95.24	94.80	-	-
8	96.79	95.34	96.50	−2	1
9	120.99	118.60	119.78	−2	1
10	140.35	138.01	131.57	−2	−5
11	141.79	138.10	138.01	−3	0
12	-	-	138.16	6	−9
13	141.85	151,15	150.27	-	-
14	154.86	153.73	-	-	-
15	-	158.87	-	-	-
16	179.06	177.29	172.81	−1	−3

**Table 3 materials-16-04722-t003:** The list of case numbers with the corresponding combinations of dimensions.

Case Number	*L*_1_ (m)	*L*_2_ (m)	*h* (mm)
0125	1.00	1.00	2.50
0150	1.00	1.00	5.00
0175	1.00	1.00	7.50
0225	1.50	1.00	2.50
0250	1.50	1.00	5.00
0275	1.50	1.00	7.50
0325	2.00	1.00	2.50
0350	2.00	1.00	5.00
0375	2.00	1.00	7.50

## Data Availability

The data presented in this study are available on request from the corresponding author.

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
