# Peer review of "Numerical Analysis, Optimization, and Multi-Criteria Design of Vacuum Insulated Glass Composite Panels"

_materials, 2023, doi:10.3390/ma16134722_

Round 1

Reviewer 1 Report

In this study, the authors present the results of designing VIG panels considering parameters such as the placement of columns in VIG panels, their quantity and spacing between glasses, as well as aesthetic reasons. The problem is new and its formulation is correct.

Two important details stand out in this study:

a) Various numerical models of VIG plates are simple and easier to implement, but give accurate results compared to advanced three-dimensional structural models.

b) Another issue is the use of multi-criteria optimization in the study.

In addition, the authors expanded the objective function with a parameter that determines the degree of similarity to the fractal in this study, to obtain a simple quantitative parameter that characterizes the aesthetic properties of the panels.

These aspects contribute to the originality of the article.

Some clarification in the article will improve the quality of the article

1) The VIG plate is used to find the vibration frequency, but the material properties of the VIG plate are not presented.

2) The vibration equation is derived, but the stress-strain relations are not presented.

3) It is not clear which theories are used to derive the fundamental equations.

4) Don't these kinds of plates have anisotropic structure?

5) Although it is not clearly stated in this study, can realistic results be obtained with linear and classical theory for frequency values?

6) In the forced vibration equation, the coefficients are written in general, what are these coefficients for VIG plates?

7) It should be checked whether it should be (1+nu) or (1-nu^2) in equation (9). etc

The article should be developed from a technical point of view.

The manuscript falls within the scope of the Journal and could be considered for publication after minor revisions.

In this study, the authors present the results of designing VIG panels considering parameters such as the placement of columns in VIG panels, their quantity and spacing between glasses, as well as aesthetic reasons. The problem is new and its formulation is correct.

Two important details stand out in this study:

a) Various numerical models of VIG plates are simple and easier to implement, but give accurate results compared to advanced three-dimensional structural models.

b) Another issue is the use of multi-criteria optimization in the study.

In addition, the authors expanded the objective function with a parameter that determines the degree of similarity to the fractal in this study, to obtain a simple quantitative parameter that characterizes the aesthetic properties of the panels.

These aspects contribute to the originality of the article.

Some clarification in the article will improve the quality of the article

1) The VIG plate is used to find the vibration frequency, but the material properties of the VIG plate are not presented.

2) The vibration equation is derived, but the stress-strain relations are not presented.

3) It is not clear which theories are used to derive the fundamental equations.

4) Don't these kinds of plates have anisotropic structure?

5) Although it is not clearly stated in this study, can realistic results be obtained with linear and classical theory for frequency values?

6) In the forced vibration equation, the coefficients are written in general, what are these coefficients for VIG plates?

7) It should be checked whether it should be (1+nu) or (1-nu^2) in equation (9). etc

The article should be developed from a technical point of view.

The manuscript falls within the scope of the Journal and could be considered for publication after minor revisions.

Reviewer 2 Report

Dear authors

The paper presented method of designing VIG composite panels through numerical, analytic, and fractal analysis. It seems that it provides useful method, but unfortunately without experimental validation. Meanwhile, the presentation of this paper should be improved. See following suggestions.

1)  The abstract should be written in a commonly used format including aim, method, results and significance. Obviously, in this study, main texts were used in describing the background. Therefore, it is recommended that the abstract should be rewritten. 

2) In Introduction, about the sentence of line 56 and 57,more literature should be  added. Following papers relates to using the fractal dimension in engineering.

     1. Characterizing Mode I Fracture Behaviors of Wood Using Compact Tension in Selected System Crack Propagation.

     2. Study on wood fracture parallel to the grains based on fractal geometry

3) In line 103, you said that "a simple quantitative parameter characterizing the aesthetic properties of panels was obtained" . ow did you characterize and evaluated the aesthetic properties of panels, which was not found in the paper?

4) In Numerical modelling, simple FEM model, 3D FEM model, and Analytical model. Some parameters were used in equations. What are values of these parameters you used in this model. Also the parameters in the FEM model.

5) About the equations cited in the text, it is recommended that All equations cited in the text should be by number instead of function .......

6) In table 1 and 2, The decimal should use "." not ",". 

7) In Fig. 10 and 11, The scale bar should be added.

Round 2

Reviewer 2 Report

Dear authors

The paper has been much revised. Hope to see your new work on this topic.